# Microbiota composition of *Culex perexiguus* mosquitoes during the West Nile virus outbreak in southern Spain

**Marta Garrigós**[1]*, **Mario Garrido**[2], **María José Ruiz-López**[1,3], **María José García-López**[4,5], **Jesús Veiga**[1], **Sergio Magallanes**[1,3], **Ramón Soriguer**[1,3], **Isabel Moreno-Indias**[4,6], **Jordi Figuerola**[1,3], **Josué Martínez-de la Puente**[1,3]

**1** Department of Conservation Biology and Global Change, Estación Biológica de Doñana (EBD), CSIC, Sevilla, Spain, **2** Department of Parasitology, Faculty of Pharmacy, University of Granada, Granada, Spain, **3** CIBER de Epidemiología y Salud Pública (CIBERESP), Madrid, Spain, **4** Department of Endocrinology and Nutrition, Instituto de Investigación Biomédica de Málaga (IBIMA), Hospital Universitario Virgen de la Victoria, Málaga, Spain, **5** Facultad de Medicina, Universidad de Málaga, Málaga, Spain, **6** CIBER de Fisiopatología de la Obesidad y Nutrición (CIBEROBN), Madrid, Spain

\* marta.garrigos@ebd.csic.es

**Data Availability Statement:** Sequence data supporting the conclusions is publicly available at https://doi.org/10.20350/digitalCSIC/16606.

## Abstract

West Nile virus (WNV) is a flavivirus naturally circulating between mosquito vectors and birds, occasionally infecting horses and humans and causing epidemiologically relevant outbreaks. In Spain, the first big WNV outbreak was recorded in 2020, resulting in 77 people infected and 8 fatalities, most of them in southern Spain. *Culex perexiguus* was identified as the primary vector of WNV maintaining its enzootic circulation of the virus. Growing evidence highlights the role of mosquito microbiota as a key component determining the vectorial capacity of mosquitoes, largely contributing to disease epidemiology. Here, we develop, to our knowledge, the first identification of the microbiota composition of this mosquito vector under natural conditions and test for the potential relationship between mosquito microbiota composition and WNV infection. To do so, we collected mosquitoes in a natural area of southern Spain during the 2020 WNV outbreak and identified the microbiota composition of mosquitoes using a 16S rRNA gene metabarcoding approach. The microbiota of *Cx. perexiguus* was dominated by the phylum *Proteobacteria*. The most abundant families were *Burkholderiaceae* and *Erwiniaceae*, including the genera *Burkholderia*, *Erwinia*, and *Pantoea*. The genus *Wolbachia*, which use to dominate the microbiota of *Cx. pipiens* and negatively interact with WNV according to the literature, had a low prevalence and relative abundance in *Cx. perexiguus* and its abundance did not differ between WNV-positive and WNV-negative mosquito pools. The microbiota diversity and composition of *Cx. perexiguus* were not significantly related to the WNV infection status. These results provide the first identification of the mosquito microbiota in an endemic area of WNV circulation in Spain.

## Introduction

West Nile virus (WNV; *Flavivirus*; *Flaviviridae*) is a significant global threat, causing disease in humans and animals worldwide. In nature, WNV is transmitted through mosquito vectors to

**Funding:** This work was supported by the MICIU/AEI/10.13039/501100011033 under grant PID2020-118205GB-I00. Additional support for this study comes from the European Commission – NextGenerationEU (Regulation EU 2020/2094), through CSIC's Global Health Platform (PTI Salud Global). M Garrido was supported by the P9 program for the Incorporation of Young Doctors funded by the University of Granada and granted by PID2022-137746NA-I00 funded by MICIU/AEI/10.13039/501100011033 and by ERDF/EU. Jesús Veiga received financial support from the Juan de la Cierva program (FJC2021-048057-I) and Marta Garrigós from the FPI program (PRE2021-098544) funded by MICIU/AEI/10.13039/501100011033 and the European Union NextGenerationEU/PRTR. MJRL was funded by the Agencia Estatal de Investigación (project PID2020-118921RJ-100/AEI/10.13039/501100011033). Isabel Moreno-Indias was supported by the "Miguel Servet Type II" program (CPII21/00013) of the ISCIII-Madrid (Spain), co-financed by the FEDER. The authors thanks for its support of the CIBER-IBIMA-Metagenomics platform, and the Genomics ECAI from IBIMA-Plataforma Bionand.

**Competing interests:** The authors have declared that no competing interests exist.

birds, its main reservoirs, and occasionally to 'dead-end' hosts like horses and humans. WNV is underdiagnosed and associated with harmful health, social, and economic consequences [1]. During recent decades, WNV caused different outbreaks in Europe, with a large increase in autochthonous infections occurring in 2018 [2]. In Spain, the local circulation of WNV has been reported since 2003 [3, 4], and the first WNV-associated disease case in humans was reported in 2004. Despite endemic circulation supported by long-term studies using wild birds [5] and feral horses [3], human cases were sporadic in subsequent years (2 cases in 2010 and 3 in 2016). In 2020, a significant outbreak occurred with 77 human cases and 8 fatalities, most of them in Seville province [6], southern Spain. Human infections have also been documented during 2024 [7], supporting the necessity to investigate the factors affecting WNV epidemiology in the area.

Mosquitoes of the *Culex* (*Cx.*) genus are the primary WNV vectors, especially the species within the *Culex pipiens* complex [1, 8]. Yet, most evidence supports the role of other species in the natural circulation of the virus. This is the case of *Culex perexiguus* which is considered a major vector in southern Spain [9]. WNV has been repeatedly identify in *Cx. perexiguus* pools captured in the area [10, 11] and the abundance of *Cx. perexiguus* has been positively associated with the prevalence of WNV in wild bird populations [12]. During the 2020 outbreak, most WNV infections were detected in *Cx. perexiguus* (33 out of 419 mosquito pools tested using Real-Time RT–PCR; 7.88%), with a much lower incidence in *Cx. pipiens* (1 out of 152 mosquito pools tested using Real-Time RT–PCR; 0.66%) and a total absence in *Cx. modestus* (75 mosquito pools tested). *Culex perexiguus* was identified as the primary vector maintaining WNV enzootic circulation in natural and agricultural areas, while *Cx. pipiens* potentially aided in the WNV transmission to humans in urban areas [8, see also 9].

Growing evidence highlights mosquito microbiota's significant role in affecting the different components of the vectorial capacity [13] and in disease transmission control [14, 15]. Mosquito microbiota of wild mosquitoes largely varies between species and areas [16, 17], and its composition may have profound effects on their vector competence and the dynamics of mosquito-borne pathogen transmission. Mosquito microbiota may directly affect the development of pathogens in mosquitoes by the competition with the pathogens for resources [18] and the secretion of anti-pathogen molecules [19], the hindrance of necessary interactions between the pathogen and vector epithelium [20] or the formation of the peritrophic matrix around the blood bolus after blood feeding, which is a barrier against pathogens [19], among other mechanisms. In addition, mosquito microbiota may modulate the immunological responses of mosquitoes finally determining pathogen development [14].

According to studies conducted under controlled conditions, WNV infection has been negatively correlated to *Wolbachia* relative abundance in *Culex* mosquitoes [21]. On the contrary, WNV infection positively correlated with the relative abundance of bacteria of the genera *Enterobacter* and *Serratia* and to bacterial diversity [21, 22]. However, despite the necessity to identify natural associations between mosquitoes and their microbiota and their effects on pathogen development, studies in the field using *Culex* mosquitoes are scarce and mainly focus on bacteria of the genus *Wolbachia*, showing no clear association patterns with WNV infections [21]. Furthermore, the microbiota of *Cx. perexiguus* and its relationship with pathogens such as WNV is virtually unknown.

Here, we characterize the bacterial community of the *Cx. perexiguus* microbiota and investigate the potential relationship between its bacterial diversity and composition and the WNV infection status. We used field-collected *Cx. perexiguus* mosquito females from a locality of the Seville province captured in August 2020, during the WNV outbreak in southern Spain, to identify the ecological interactions occurring under natural conditions.

## Material and methods

### Mosquito collection

Mosquitoes were collected on 27th of August 2020 in La Dehesa de Abajo (Seville, southern Spain; 6˚14'W, 36˚57'N), close to the area where the main WNV human outbreak occurred in 2020. La Dehesa de Abajo is a natural protected area surrounded by rice fields which include a lagoon hosting a diversity of bird species. Mosquitoes were captured using Biogents (BG)-sentinel-2 traps (Biogents, Regensbourg, Germany) supplemented with $CO_2$. Trapping of mosquitoes was done with all the necessary permits from landowners and the local authorities (Consejería de Medio Ambiente, Junta de Andalucía). Trapped mosquitoes were sexed and identified at species level using available morphological keys [23–25]. A total of 1,000 *Cx. perexiguus* females, the only hematophagous sex, with no signs of recent blood ingestion (to avoid virus amplification from bloodmeals) were grouped into 100 pools of 10 mosquitoes each and maintained frozen (-80˚C) until molecular analyses.

### Molecular procedures

RNA and DNA were simultaneously extracted from each mosquito pool with the Maxwell® extraction robot and the Viral Total Nucleic Acid Purification kit (Promega, Madison, Wisconsin, USA), following manufacturer instructions. This procedure allows the extraction of both RNA and DNA from the samples. RNA was subsequently used to screen for WNV infection and DNA for mosquito microbiota analyses. The surface of mosquitoes was not sterilized before the extraction of genetic material to avoid RNA degradation in the samples. Non-significant differences have been previously found in the bacterial community of surface sterilized and non-sterilized insects [26].

The WNV infection status was tested using a RT–PCR protocol that amplifies all the known WNV lineages [27]. WNV-positive samples (n = 19) and a comparable number of WNV-negative mosquito pools (n = 21) were used for the molecular characterization of the bacterial microbiota of mosquitoes. In these samples, DNA concentration and purity were estimated with a Nanodrop spectrophotometer (Nanodrop Technologies, Wilmington, Delaware, USA). For microbiota analyses, libraries from each mosquito pool were built with the Ion 16S Metagenomics kit (Thermofisher, Waltham, Massachusetts, USA), consisting of primer pools to amplify multiple variable regions (V2, 3, 4, 6–7, 8 and 9) of the 16S rRNA. After generating amplicons, the Ion PlusTM Fragment Library Kit (Thermofisher, Waltham, Massachusetts, USA) was used to ligate barcoded adapters and synthesize libraries. Barcoded libraries from all the samples were pooled and templated on the automated Ion Chef system (Thermofisher, Waltham, Massachusetts, USA) followed by a 400 bp sequencing on the Ion S5 (Thermofisher, Waltham, Massachusetts, USA).

The quality of the reads was checked using FastQC (ver. 0.12.1) [28] and MultiQC (ver. 1.17) [29]. Bacteria sequences obtained from mosquito pools were translated into amplicon sequence variants (ASVs) using DADA2 [30] with the microbiome analysis package QIIME2 (ver. 2023.5.1) [31]. QIIME2 was also used to assign each ASV to a taxonomic group with an identity of 99% based on the SILVA database (ver. 138.1) [32]. Subsequently, reads were filtered removing singletons, reads with low taxonomic resolution (below phylum), and non-bacterial, chloroplast, and mitochondrial ASVs. Furthermore, rarefaction curves were plotted to ensure that all samples had enough sequencing depth to capture the bacterial community diversity (See S1 File for the script of QIIME2 analysis).

## Statistical analyses

Statistical analyses and graphical representations were carried out in R [33] with the Bioconductor packages *qiime2R* (ver. 0.99.6) [34], *phyloseq* (ver. 1.38.0) [35], *microViz* (ver.0.11.0) [36], and *microbiome* (ver. 1.16.0) [37]. WNV prevalence was calculated using Epitools (https://epitools.ausvet.com.au) considering mosquito pools of equal size and assuming 100% sensitivity and specificity. The most abundant taxa were described from phylum to genus levels. In addition, we assessed the prevalence and relative abundance of the genera *Wolbachia*, *Serratia*, and *Enterobacter* because of their previously reported correlation with WNV infection in *Culex* mosquitoes [16, 22]. Most statistical analyses were performed at the family level as it provided the greatest certainty about taxonomic identification. ASVs with a taxonomic resolution below family or ambiguous family annotation were not included in the analyses. Bacterial alpha diversity (within-sample diversity) was estimated through observed richness and Shannon index. The Shannon index takes into account both species richness and evenness within a sample, being higher when the number of taxa is higher and more evenly distributed. Beta-diversity (between-sample diversity) was explored by ordination analyses including a Principal Coordinate Analysis (PCoA) of both Bray-Curtis dissimilarity matrix for family abundances and Jaccard similarity matrix for family presence/absence. The relationship between mosquito microbiota and WNV infection was assessed using linear models (LMs) to test if the observed richness or the Shannon index varied according to the WNV infection status. To assess the association between mosquito microbiota beta diversity and WNV infection, we performed a PERmutational Multivariate ANalysis Of VAriance (PERMANOVA) to test whether PCoA ordination analyses for Bray-Curtis dissimilarity and Jaccard similarity matrices clustered samples by WNV infection status. Statistical significance of F-values obtained from PERMANOVA was determined by comparison to 999,999 permutations. Differences in the microbiota composition in mosquito pools according to the WNV infection were checked for all taxonomic levels from phylum to species when available. We first filtered taxa with a minimum prevalence of 10% (to avoid spurious ASVs) and then fitted a LM with a log2-transformed response including the relative abundance of all taxa found in the filtered dataset as the dependent variable and the WNV infection status as the independent variable. The Benjamini-Hochberg procedure was used to adjust the p-values for multiple testing. The level WNV-negative was set as the reference level in all LMs including the WNV infection status as the independent variable (the R script is shown in S2 File to R script).

## Results

Overall, 1,000 *Cx. perexiguus* females grouped in 100 pools of 10 mosquitoes each were molecularly tested for the presence of WNV. Of them, 19 mosquito pools were positive (19%), representing an estimated prevalence of 0.02 (Confidence limits 2.5%: 0.01; 97.5%: 0.03; SE: 0.0047). These 19 WNV-positive mosquito pools together with 21 WNV-negative mosquito pools were used for the subsequent analyses.

### Characterization of *Cx. perexiguus* microbiota

We analyzed the microbiota of 40 *Cx. perexiguus* pools obtaining a total of 6,453,096 reads, ranging from 46,513 to 419,497 (average = 161,327.4). The rarefaction curves reached saturation for all samples, indicating that our sampling effort captured most of the bacterial diversity (S1 Fig). We identified a relatively rich bacterial community in *Cx. perexiguus* pools with a total of 4,434 ASVs belonging to 18 phyla, 33 classes, 85 orders, 168 families, and 410 genera.

The microbiota of *Cx. perexiguus* mosquito pools was dominated by the phylum *Proteobacteria*. The *Burkholderiaceae* family was the most abundant (average relative abundance = 0.676,

SD = 0.150) except in two samples. In these two samples, the *Erwiniaceae* family was the most abundant (average relative abundance = 0.085, SD = 0.137). Following these two families, the next most abundant families in mosquito pools were *Rhodobacteraceae*, *Cellulomonadaceae*, *Micrococcaceae*, *Microbacteriaceae*, and *Enterobacteriaceae* (Fig 1A; S2 Fig). *Burkholderia* (*Burkholderiaceae*) was the most abundant genus in mosquito pools (average relative abundance = 0.696, SD = 0.152), followed by *Erwinia* and *Pantoea* (*Erwiniaceae*) with a relative abundance of 0.050 (SD = 0.091) and 0.037 (SD = 0.093), respectively. The genera *Wolbachia*, *Serratia*, and *Enterobacter* were identified in 4, 4 and 33 mosquito pools, with a maximum relative abundance of <0.001, 0.002, and 0.004, respectively.

### Relationships between *Cx. perexiguus* microbiota and WNV infection

We did not find any significant difference in the alpha and beta diversity (Fig 1B and 1C; S3 and S4 Figs) of *Cx. perexiguus* pools according to the WNV infection status (all P > 0.05). Likewise, we did not detect significant differences in the relative abundances of any taxa, at any taxonomic level, between mosquito pools based on their infection status (all P > 0.05). This was also true for analyses performed for bacteria of the genera *Wolbachia*, *Serratia*, and *Enterobacter*.

## Discussion

Mosquito microbiota has been highlighted as a major driver of vector competence by affecting the development of pathogens directly and indirectly, supporting the importance of studying their composition in natural mosquito populations. To our knowledge, here we characterized for the first time the microbiota composition of field-collected *Cx. perexiguus* mosquito females captured in an area of endemic circulation of vector-borne pathogens, including the zoonotic WNV.

Using a metabarcoding approach, we identified a relatively rich bacterial community in female *Cx. perexiguus* pools representing 410 different genera. Among them, bacteria of the genera *Burkholderia* (*Burkholderiaceae*), *Erwinia* and *Pantoea* (*Erwiniaceae*) dominated, in terms of relative abundance, the microbiota of *Cx. perexiguus* females. These genera are commonly found in *Aedes* spp. and *Anopheles* spp. mosquitoes, both in the wild and laboratory colonies [38]. However, they are rare in the microbiota of *Culex* mosquitoes [39]. In addition, *Wolbachia* showed a low prevalence in the studied mosquito population. While *Cx. pipiens* has a very high prevalence of *Wolbachia* in different populations [40], other related species of the *Culex* genus may show an opposite pattern [40, 41]. This is the case of species including *Culex torrentum* [41], *Culex restuans* [42] or *Cx. perexiguus*, as shown here. These results suggest that the microbiota composition of mosquitoes may largely differ between species, including those breeding in the same area [41, 42]. However, our results are limited by the fact that we only sampled mosquitoes a single population and time point, thus further studies are necessary to identify the natural variation of the microbiota composition of this species in a broader study area.

Mosquito microbiota has been highlighted as a major driver of vector competence by affecting the development of pathogens both directly and indirectly. Different field and laboratory studies on *Culex* spp. have associated WNV infection with changes in the microbiota community composition of mosquitoes, including changes in the abundance of the most prevalent microbiota genera [21]. For instance, Zink *et al*. [22] exposed *Cx. pipiens* mosquitoes to WNV and observed that WNV-exposed and infected mosquitoes showed lower relative abundance of *Wolbachia* but higher relative abundance of bacteria of the genera *Enterobacter* and *Serratia* and bacterial diversity. In this study, authors also observed an up-regulation of numerous

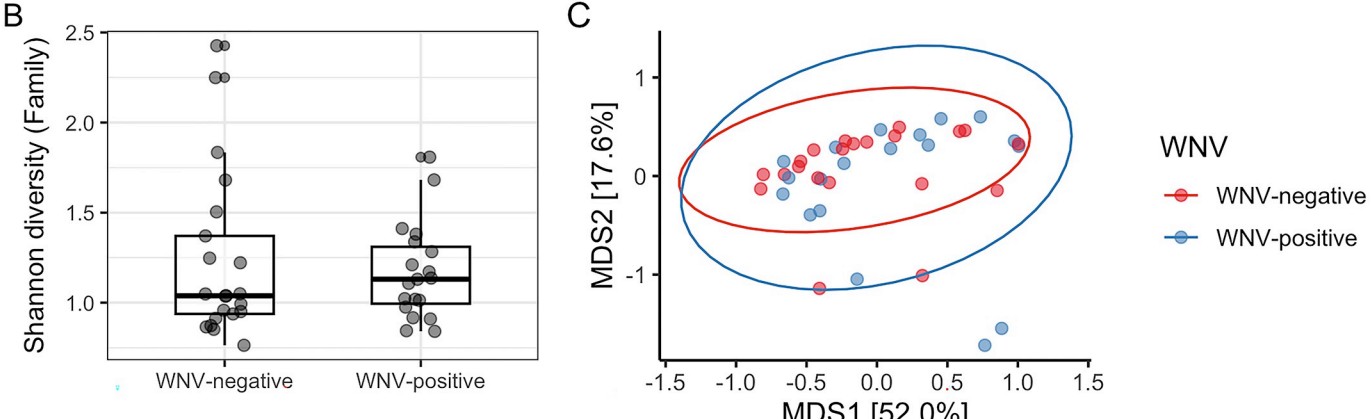

**Fig 1. A:** Relative abundance of the 12 most abundant taxa at family level in negative and positive *Cx. perexiguus* mosquito pools for West Nile virus (WNV). **B:** Distribution of the Shannon diversity index (alpha-diversity) at the family level of *Cx. perexiguus* mosquito pools according to the infection status by WNV. The vertical lines go from the lower and upper quartiles to the minimum or maximum, respectively, and the horizontal line represents the median. **C:** Principal Coordinates Analysis (PCoA) for Bray-Curtis distance matrix (relative abundance) by WNV infection at family level. Percentages shown in MDS1 and MDS2 axes refer to the percentage of variation explained by each of the two selected main coordinate axes.

genes related to the mosquito innate immune system, suggesting an indirect interaction between WNV and the mosquito microbiome through the immune response. On the other hand, *Wolbachia* is known to block replication of WNV and other viruses such as dengue, Chikungunya, and West Nile virus [18, 43, 44], while its absence may affect the susceptibility to pathogen infection by mosquitoes. For example, *Wolbachia*-free *Cx. modestus* mosquitoes are major vectors of WNV in endemic regions, relegating sympatric *Cx. pipiens* to a secondary role [41, 45]. Likewise, according to our results, *Cx. perexiguus*, the primary vector of WNV during the 2020 Spanish outbreak [9], exhibits a low prevalence of *Wolbachia* potentially explaining, at least in part, the role of this species in the transmission of WNV in the area. In addition, the low presence of this taxa in *Cx. perexiguus* could potentially explain the absence of differences in the microbiota composition of mosquitoes according to their WNV infection status. Further studies using *Cx. perexiguus* mosquitoes reared under controlled conditions could help to assess the link between mosquito microbiota and WNV-infection in wild mosquitoes in the future. These studies may consider using *Wolbachia* bacteria (strain *w*Mel), which reduce the lifespan and partially block viral infections in other mosquito species such as *Aedes aegypti* [18, 46].

## Conclusion

This study provides the first information about the microbiome composition of *Cx. perexiguus*, a major WNV vector in southern Spain, during the largest WNV outbreak ever recorded in the country. The low prevalence of *Wolbachia* in *Cx. perexiguus* mosquitoes could potentially explain, at least in part, the relevance of this mosquito species in the WNV transmission and makes necessary experimental tests to identify the impact of mosquito microbiota composition on *Cx. perexiguus* vector capacity. However, in order to conduct these studies, previous information about the composition of the microbiota of mosquitoes in the wild is essential.

## Supporting information

**S1 Fig. Rarefaction curves of the microbiota analyses of each *Cx. perexiguus* mosquito pool included in the study.** Rarefaction of the samples consists in randomly keep a specific number of sequencing reads from the sample, removing the rest of the reads. The rarefaction curve represents the number of ASVs present in each sample (y-axis) when rarified to different number of reads (x-axis). If a sample plateau, it indicates that its sequencing depth was sufficient to represent the bacterial diversity in that sample.
(PNG)

**S2 Fig. Heatmap of Centered Log Ratio (CLR) transformation of taxa relative abundance at family level.** The figure shows the 15 most abundant families. Higher CLR values are colored red and correspond to higher relative abundances, while lower CLR values are colored blue and correspond to lower relative abundances. In the legend above the graph blue corresponds to WNV-positive samples and light blue to WNV-negative samples. The tree below the graph shows the samples grouped according to the similarity of microbiota composition based on Euclidean distances.
(PDF)

**S3 Fig. Distribution of the microbiota richness at the family level of *Cx. perexiguus* mosquito pools according to the infection status by WNV.** The vertical lines go from the lower and upper quartiles to the minimum or maximum, respectively, and the horizontal line represents the median.
(PDF)

**S4 Fig. Principal Co-ordinates Analysis (PCoA) for Jaccard matrix of the microbiota *Cx. perexiguus* mosquito pools at family level according to the infection status by WNV.** The percentage of variation explained by each component (axis) is shown in square brackets. (PDF)

**S1 File. QIIME2 pipeline.**
(DOCX)

**S2 File. R script.**
(DOCX)

## Acknowledgments

We thank the contribution of Álvaro Solís during the fieldwork. Two anonymous reviewers provided valuable comments on a previous version of the manuscript

## Author Contributions

**Conceptualization:** Jordi Figuerola, Josué Martínez-de la Puente.

**Data curation:** Marta Garrigós.

**Formal analysis:** Marta Garrigós, María José Ruiz-López, María José García-López, Jesús Veiga, Isabel Moreno-Indias.

**Funding acquisition:** Jordi Figuerola, Josué Martínez-de la Puente.

**Investigation:** Marta Garrigós, María José Ruiz-López, Jesús Veiga, Sergio Magallanes, Ramón Soriguer, Isabel Moreno-Indias, Jordi Figuerola, Josué Martínez-de la Puente.

**Methodology:** Marta Garrigós, María José Ruiz-López, Jesús Veiga, Isabel Moreno-Indias, Jordi Figuerola, Josué Martínez-de la Puente.

**Project administration:** Josué Martínez-de la Puente.

**Resources:** Josué Martínez-de la Puente.

**Software:** Marta Garrigós, Jesús Veiga.

**Supervision:** Josué Martínez-de la Puente.

**Validation:** Jordi Figuerola, Josué Martínez-de la Puente.

**Visualization:** Marta Garrigós, Mario Garrido.

**Writing – original draft:** Marta Garrigós, Mario Garrido.

**Writing – review & editing:** Marta Garrigós, Mario Garrido, María José Ruiz-López, María José García-López, Jesús Veiga, Sergio Magallanes, Ramón Soriguer, Isabel Moreno-Indias, Jordi Figuerola, Josué Martínez-de la Puente.

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
