## [Decision Letter · Decision Letter 0]

4 Oct 2024

PONE-D-24-36152Microbiota composition of Culex perexiguus mosquitoes during the West Nile virus outbreak in southern SpainPLOS ONE

Dear Dr. Garrigós,

Thank you for submitting your manuscript to PLOS ONE. After careful consideration, we feel that it has merit but does not fully meet PLOS ONE’s publication criteria as it currently stands. Therefore, we invite you to submit a revised version of the manuscript that addresses the points raised during the review process.

Both reviewers have identified items that should be reviewed and discussed in order to increase value and relevance of this really great study.

We look forward to receiving your revised manuscript.

Kind regards,

Kelli L. Barr, Ph.D.

Academic Editor

PLOS ONE

Journal Requirements:

“This work was supported by the MICIU/AEI/10.13039/501100011033 under grant PID2020-118205GB-I00. Additional support for this study comes from the European Commission – NextGenerationEU (Regulation EU 2020/2094), through CSIC’s Global Health Platform (PTI Salud Global). M Garrido was supported by the P9 program for the Incorporation of Young Doctors funded by the University of Granada and granted by PID2022-137746NA-I00 funded by MICIU/AEI/10.13039/501100011033 and by ERDF/EU. Jesús Veiga received financial support from the Juan de la Cierva program (FJC2021-048057-I) and Marta Garrigós from the FPI program (PRE2021-098544) funded by MICIU/AEI/10.13039/501100011033 and the European Union NextGenerationEU/PRTR. MJRL was funded by the Agencia Estatal de Investigación (project PID2020-118921RJ-100/AEI/10.13039/501100011033). Isabel Moreno-Indias was supported by the "Miguel Servet Type II" program (CPII21/00013) of the ISCIII-Madrid (Spain), co-financed by the FEDER. The authors thanks for its support of the CIBER-IBIMA-Metagenomics platform, and the Genomics ECAI from IBIMA-Plataforma Bionand.”

“We thank the contribution of Álvaro Solís during the fieldwork. This work was supported by the MICIU/AEI/10.13039/501100011033 under grant PID2020-118205GB-I00. Additional support for this study comes from the European Commission – NextGenerationEU (Regulation EU 2020/2094), through CSIC’s Global Health Platform (PTI Salud Global). M Garrido was supported by the P9 program for the Incorporation of Young Doctors funded by the University of Granada and granted by PID2022-137746NA-I00 funded by MICIU/AEI/10.13039/501100011033 and by ERDF/EU. Jesús Veiga received financial support from the Juan de la Cierva program (FJC2021-048057-I) and Marta Garrigós from the FPI program (PRE2021-098544) funded by MICIU/AEI/10.13039/501100011033 and the European Union NextGenerationEU/PRTR. MJRL was funded by the Agencia Estatal de Investigación (project PID2020-118921RJ-100/AEI/10.13039/501100011033). Isabel Moreno-Indias was supported by the "Miguel Servet Type II" program (CPII21/00013) of the ISCIII-Madrid (Spain), co-financed by the FEDER. The authors thanks for its support of the CIBER-IBIMA-Metagenomics platform, and the Genomics ECAI from IBIMA-Plataforma Bionand.”

“This work was supported by the MICIU/AEI/10.13039/501100011033 under grant PID2020-118205GB-I00. Additional support for this study comes from the European Commission – NextGenerationEU (Regulation EU 2020/2094), through CSIC’s Global Health Platform (PTI Salud Global). M Garrido was supported by the P9 program for the Incorporation of Young Doctors funded by the University of Granada and granted by PID2022-137746NA-I00 funded by MICIU/AEI/10.13039/501100011033 and by ERDF/EU. Jesús Veiga received financial support from the Juan de la Cierva program (FJC2021-048057-I) and Marta Garrigós from the FPI program (PRE2021-098544) funded by MICIU/AEI/10.13039/501100011033 and the European Union NextGenerationEU/PRTR. MJRL was funded by the Agencia Estatal de Investigación (project PID2020-118921RJ-100/AEI/10.13039/501100011033). Isabel Moreno-Indias was supported by the "Miguel Servet Type II" program (CPII21/00013) of the ISCIII-Madrid (Spain), co-financed by the FEDER. The authors thanks for its support of the CIBER-IBIMA-Metagenomics platform, and the Genomics ECAI from IBIMA-Plataforma Bionand.”

6. Please respond by return e-mail with an updated version of your manuscript to amend either the abstract on the online submission form or the abstract in the manuscript so that they are identical. We can make any changes on your behalf.

Reviewers' comments:

Reviewer's Responses to Questions

**Comments to the Author**

1. Is the manuscript technically sound, and do the data support the conclusions?

Reviewer #1: Yes

Reviewer #2: Partly

2. Has the statistical analysis been performed appropriately and rigorously? 

Reviewer #1: Yes

Reviewer #2: Yes

3. Have the authors made all data underlying the findings in their manuscript fully available?

Reviewer #1: Yes

Reviewer #2: Yes

4. Is the manuscript presented in an intelligible fashion and written in standard English?

Reviewer #1: Yes

Reviewer #2: Yes

5. Review Comments to the Author

Reviewer #1: Interesting and well-conducted study.

1. The abstract can contain more findings from the molecular analyses (abstracts are different in the system and manuscript).

2. Please interpret more on the bacterial diversity using the Shannon Index.

3. Can include discussion on the potential of Wolbachia as a vector/disease control tool for WNV transmission.

4. Make the rarefaction curve self-explanatory.

Reviewer #2: This research represents the first investigation into the composition of the Cx. Perexiguus microbiota, marking a valuable contribution to the field. However, the findings would be further enriched by the availability of samples collected from multiple locations and at various times.

The results indicate no discernible difference in microbiota composition between WNV-infected and non-infected mosquitoes, nor in their Wolbachia infection rates. Interestingly, the Wolbachia infection rate in Cx. Perexiguus is notably low, potentially increasing susceptibility to WNV infection. The interaction between WNV and Wolbachia infections, however, remains unclear and warrants further exploration.

6. PLOS authors have the option to publish the peer review history of their article (what does this mean?). If published, this will include your full peer review and any attached files.

Reviewer #1: No

Reviewer #2: No

---

## [Author Response · Author response to Decision Letter 0]

22 Oct 2024

Dear Editor,

here we enclosed the revised version of the manuscript entitled “Microbiota composition of Culex perexiguus mosquitoes during the West Nile virus outbreak in southern Spain” for its consideration for publication in PLOS ONE. We have carefully revised the text to include all the comments and suggestions proposed by the reviewers and the editor point by point (changes are highlighted in yellow). We thank their valuable contribution, which has significantly improved the previous version of the manuscript. We hope that following these changes the new version of the manuscript is acceptable for publication in PLOS ONE.

Looking forward to your notices,

Sincerely,

Marta Garrigós

Please see the responses to the Editor’s comments:

ANSWER: we have ensured that the manuscript meets PLOS ONE's style requirements. 

ANSWER: Mosquito sampling was authorized by the Junta de Andalucía. We have included this information in the new version of the manuscript. Mosquitoes are not protected by any law, so additional animal experimentation authorizations are not necessary. Please, see lines 106-108.

ANSWER: We corrected the funding details in this submission. This information is not included in the main text but in the submission platform.

ANSWER: The funders had no role in study design, data collection and analysis, decision to publish, or preparation of the manuscript. The corrected financial disclosure is:

“This work was supported by the MICIU/AEI/10.13039/501100011033 under grant PID2020-118205GB-I00. Additional support for this study comes from the European Commission – NextGenerationEU (Regulation EU 2020/2094), through CSIC’s Global Health Platform (PTI Salud Global). M Garrido was supported by the P9 program for the Incorporation of Young Doctors funded by the University of Granada and granted by PID2022-137746NA-I00 funded by MICIU/AEI/10.13039/501100011033 and by ERDF/EU. Jesús Veiga received financial support from the Juan de la Cierva program (FJC2021-048057-I) and Marta Garrigós from the FPI program (PRE2021-098544) funded by MICIU/AEI/10.13039/501100011033 and the European Union NextGenerationEU/PRTR. MJRL was funded by the Agencia Estatal de Investigación (project PID2020-118921RJ-100/AEI/10.13039/501100011033). Isabel Moreno-Indias was supported by the "Miguel Servet Type II" program (CPII21/00013) of the ISCIII-Madrid (Spain), co-financed by the FEDER. The authors thanks for its support of the CIBER-IBIMA-Metagenomics platform, and the Genomics ECAI from IBIMA-Plataforma Bionand. The funders had no role in study design, data collection and analysis, decision to publish, or preparation of the manuscript.”

ANSWER: We removed funding information from the Acknowledgments section and corrected the funding details in the submission platform. The corrected statement in Acknowledgements is:

“We thank the contribution of Álvaro Solís during the fieldwork. Two anonymous reviewers provided valuable comments on a previous version of the manuscript” (Lines 283-284)

6. Please respond by return e-mail with an updated version of your manuscript to amend either the abstract on the online submission form or the abstract in the manuscript so that they are identical. We can make any changes on your behalf.

ANSWER: We have responded by return e-mail with an updated version of the manuscript with the definitive abstract, that has been modified following the reviewers’ comments (lines 30-36). 

ANSWER: We have included the captions for Supporting Information files at the end of the manuscript (Lines 414-436). 

ANSWER: We have ensured that the reference list is complete and correct. We have changed the article cited in the line 264 as we consider that the new reference better supports the corresponding statement. To our knowledge, none of the cited paper have been retracted. In addition, we have added one reference following the reviewers’ comments (line 272)

Please, see below the responses to the queries proposed by reviewers.

Reviewer #1: Interesting and well-conducted study.

1. The abstract can contain more findings from the molecular analyses (abstracts are different in the system and manuscript).

ANSWER: we have added to the abstract the most abundant bacteria found at phylum and family level (lines 30-32) and the absence of significant differences in Wolbachia abundance between WNV-positive and WNV-negative samples (lines 35-36). In addition, we have carefully revised both the abstract in the submission platform and the manuscript which are now the same. Thanks for your comment.

2. Please interpret more on the bacterial diversity using the Shannon Index.

ANSWER: We have included a sentence in the Methods section to clarify how the Shannon index should be interpreted (lines 159-160).

3. Can include discussion on the potential of Wolbachia as a vector/disease control tool for WNV transmission.

ANSWER: We included new sentences at the end of the manuscript following your suggestion. The sentences included are: “Further studies using Cx. perexiguus mosquitoes reared under controlled conditions could be a reasonable approach to assess the relationship between mosquito microbiota and WNV-infection in wild mosquitoes in the future. These studies may consider the use of Wolbachia bacteria (strain wMel), which reduce the lifespan and partially block viral infections in other mosquito species such as Aedes aegypti.”. Please, see lines (268-272). 

4. Make the rarefaction curve self-explanatory.

ANSWER: We have added a legend to S1 Fig to make it self-explanatory (lines 415-420).

Reviewer #2: This research represents the first investigation into the composition of the Cx. Perexiguus microbiota, marking a valuable contribution to the field. However, the findings would be further enriched by the availability of samples collected from multiple locations and at various times.

ANSWER: We are deeply grateful to the reviewer for recognizing the importance of our work. We are aware of the limitations of the study in terms of the lack of different time and spatial points, as indicated in the Discussion section (please, see lines 244-247). Please note that our study is limited by the need to have a very high prevalence of West Nile in the field captured mosquitoes and this only occurs at a few localities at the moment of maximum virus transmission.

The results indicate no discernible difference in microbiota composition between WNV-infected and non-infected mosquitoes, nor in their Wolbachia infection rates. Interestingly, the Wolbachia infection rate in Cx. Perexiguus is notably low, potentially increasing susceptibility to WNV infection. The interaction between WNV and Wolbachia infections, however, remains unclear and warrants further exploration.

ANSWER: We agree that further research is needed to clarify the relationship between WNV and Wolbachia infections. We have added a sentence to the Discussion section in this regard (lines 268-274). We have also included some lines to discuss the potential relevance of Wolbachia affecting the transmission success of viruses by mosquitoes (please, see comments of Reviewer 1).

---

## [Decision Letter · Decision Letter 1]

5 Nov 2024

Microbiota composition of Culex perexiguus mosquitoes during the West Nile virus outbreak in southern Spain

PONE-D-24-36152R1

Dear Dr. Garrigós,

We’re pleased to inform you that your manuscript has been judged scientifically suitable for publication and will be formally accepted for publication once it meets all outstanding technical requirements.

Kind regards,

Kelli L. Barr, Ph.D.

Academic Editor

PLOS ONE

Additional Editor Comments (optional): While reviewer 2 was unable to address your revisions, I feel that their comments were addressed sufficiently in the revisions.

Reviewers' comments:

Reviewer's Responses to Questions

**Comments to the Author**

1. If the authors have adequately addressed your comments raised in a previous round of review and you feel that this manuscript is now acceptable for publication, you may indicate that here to bypass the “Comments to the Author” section, enter your conflict of interest statement in the “Confidential to Editor” section, and submit your "Accept" recommendation.

Reviewer #1: All comments have been addressed

2. Is the manuscript technically sound, and do the data support the conclusions?

Reviewer #1: Yes

3. Has the statistical analysis been performed appropriately and rigorously? 

Reviewer #1: Yes

4. Have the authors made all data underlying the findings in their manuscript fully available?

Reviewer #1: Yes

5. Is the manuscript presented in an intelligible fashion and written in standard English?

Reviewer #1: Yes

6. Review Comments to the Author

Reviewer #1: (No Response)

7. PLOS authors have the option to publish the peer review history of their article (what does this mean?). If published, this will include your full peer review and any attached files.

Reviewer #1: No

---

## [Editor Report · Acceptance letter]

7 Nov 2024

PONE-D-24-36152R1 

PLOS ONE

Dear Dr. Garrigós, 

I'm pleased to inform you that your manuscript has been deemed suitable for publication in PLOS ONE. Congratulations! Your manuscript is now being handed over to our production team.

Kind regards, 

on behalf of

Dr. Kelli L. Barr 

Academic Editor

PLOS ONE